# Usefulness of ^18^F-FDG PET/CT in Patients with Cardiac Implantable Electronic Device Suspected of Late Infection

**DOI:** 10.3390/jcm9072246

**Published:** 2020-07-15

**Authors:** Giuseppe Rubini, Cristina Ferrari, Domenico Carretta, Luigi Santacroce, Rossella Ruta, Francesca Iuele, Valentina Lavelli, Nunzio Merenda, Carlo D’Agostino, Angela Sardaro, Artor Niccoli Asabella

**Affiliations:** 1Nuclear Medicine Unit, Interdisciplinary Department of Medicine – University of Bari “Aldo Moro”, Piazza Giulio Cesare, 11, 70124 Bari, Italy; giuseppe.rubini@uniba.it (G.R.); ferrari_cristina@inwind.it (C.F.); rossella.ruta@yahoo.it (R.R.); francescaiuele@hotmail.com (F.I.); valentina.lavelli@gmail.com (V.L.); nu.me@hotmail.it (N.M.); 2CardioThoracic Department – Policlinic of Bari, Piazza Giulio Cesare, 11, 70124, Bari, Italy; carrettacardiologia@gmail.com (D.C.); carlo.dagostino@policlinico.ba.it (C.D.); 3Ionian Department, Microbiology and Virology Lab – University of Bari “Aldo Moro”, Piazza Giulio Cesare, 11, 70124 Bari, Italy; luigi.santacroce@uniba.it; 4Radiation Oncology Unit, Interdisciplinary Department of Medicine – University of Bari “Aldo Moro”, Piazza Giulio Cesare, 11, 70124 Bari, Italy; 5Nuclear Medicine Unit, A. O. Policlinic “A. Perrino”, Strada Statale 7 per Mesagne, 72100 Brindisi, Italy; artor.niccoliasabella@asl.brindisi.it

**Keywords:** ^18^F-FDG PET/CT, infection, cardiac implantable electronic device

## Abstract

The presence of a cardiovascular implantable electronic device (CIED) can be burdened by complications such as late infections that are associated with significant morbidity and mortality and require immediate and effective treatment. The aim of this study was to evaluate the role of ^18^F-fluorodeoxyglucose positron-emission tomography/computed tomography (^18^F-FDG PET/CT) in patients with suspected CIED infection. Fifteen patients who performed a ^18^F-FDG PET/CT for suspicion of CIED infection were retrospectively analyzed; 15 patients, with CIED, that underwent ^18^F-FDG PET/CT for oncological reasons, were also evaluated. Visual qualitative analysis and semi-quantitative analysis were performed. All patients underwent standard clinical management regardless ^18^F-FDG PET/CT results. Sensitivity, specificity, accuracy, positive predictive value (PPV) and negative predictive value (NPV) resulted as 90.91%, 75%, 86.67%, 90.91% and 75% respectively. Maximum standardized uptake values (SUV_max_) and semi-quantitative ratio (SQR) were collected and showed differences statistically significant between CIED infected patients and those who were not. Exploratory cut-off values were derived from receiver operating characteristic (ROC) curves for SUV_max_ (2.56) and SQR (4.15). This study suggests the clinical usefulness of ^18^F-FDG PET/CT in patients with CIED infection due to its high sensitivity, repeatability and non-invasiveness. It can help the clinicians in decision making, especially in patients with doubtful clinical presentation. Future large-scale and multicentric studies should be conducted to establish precise protocols about ^18^F-FDG PET/CT performance.

## 1. Introduction

Implantations of cardiac implantable electronic devices (CIEDs) have increased significantly over recent years, due to growing evidence of improved quality of life, population growth and increased life expectancy [1,2,3]. Despite the many benefits of this surgical practice, it can be burdened by complications such as infections that are associated with significant morbidity and mortality and require immediate and effective treatment [4,5,6,7]. CIED infections (CIEDIs) can onset late after placement and in these cases the diagnosis is more difficult because presentation is highly variable; a significant number of late infections presents with more indolent manifestation [8,9,10]. Delays in diagnosing and treating can result in progression to infectious endocarditis or sever sepsis with worse clinical outcomes [9,11]. CIED consists on both intravascular and extravascular components and any part of it can be involved by infection. Once any segment of the device gets involved by infection, the entire system is considered infected [12,13]. The main therapeutic option is complete device removal, which is complex, expensive and potentially accompanied by complications [13]. For this reason it is important to have as much information as possible to help clinicians choosing the most appropriate treatment [14]. ^18^F-fluorodeoxyglucose positron-emission tomography/computed tomography (^18^F-FDG PET/CT) is a validated multimodality whole-body technique that can identify invective foci because ^18^F-FDG uptake increases due to the high concentration of neutrophils and monocyte/macrophages, which overexpress glucose transporters, and hexokinase activity. For this reason ^18^F-FDG PET/CT has been recently proposed also in the diagnostic workflow of numerous infectious conditions [15,16,17,18,19].

The aim of this study was to investigate the possible role of ^18^F-FDG PET/CT in the diagnosis of suspected CIED infection.

## 2. Materials and Methods

### 2.1. Design and Patients

This observational and retrospective analysis included 30 patients with CIED implanted at least 6 months before the performance of ^18^F-FDG PET/CT. Written informed consent for collecting data for clinical research was obtained from all patients at the moment of the first hospital admission. ^18^F-FDG PET/CT were performed from November 2017 to December 2018. Our institutional review board did not require ethical committee approval for the review of patients’ files and data. 15/30 patients (14 men and 1 woman, mean age 69 years, range: 46–84 years) performed ^18^F-FDG PET/CT for suspected CIED infection (group CIEDIsusp) [12]. The suspicion of CIED infection was postulated according to the presence of at least 2 of the following signs: (1) clinical signs: fever >38 °C, local signs of generator pocket infection (erythema and/or localized cellulitis and/or swelling and/or discharge and/or dehiscence and/or pain over the pocket and/or fluid collection and/or CIED exposure); (2) laboratory signs: increased values of inflammatory index: erythrosedimentation rate (ESR) and/or C-reactive protein (CRP) and/or procalcitonin (PCT) and/or white blood cells (WBC), blood culture positivity; (3) instrumental signs: trans-thoracic echocardiography (TTE) positivity, trans-esophageal echocardiography (TEE) positivity. 15/30 patients (13 men and 2 women, mean age 76 years, range: 59–93 years) were selected as control group among patients with CIED who underwent ^18^F-FDG PET/CT for oncological surveillance without clinical suspicion of CIED infection (group ONCOctrl).

### 2.2. ^18^F-Fluorodeoxyglucose Positron-Emission Tomography/Computed Tomography (^18^F-FGD PET/CT) Technique

All patients were instructed to fast for 8 h before the exam; CIEDIsusp patients also underwent a fat-enriched and lacking carbohydrates diet for 24 h before the 8-h fast, in order to reduce the physiological uptake of the ^18^F-FDG by myocardium [20]. Patients’ blood glucose level was evaluated before ^18^F-FDG administration and all patients had a capillary level lower than 150 mg/dL. Images were acquired with a combined modality PET/CT Discovery LSA (GE Healthcare, Waukesha, Wisconsin, USA), integrating a PET with a 16-slice low-dose CT scanner, in order to perform PET images’ correction for attenuation and anatomical reconstruction. The image acquisition was obtained 45–60 min after the intravenous injection of a dose of 2.5–3.0 MBq/kg of ^18^F-FDG. Patients were hydrated by drinking 500 mL of water and instructed to empty the bladder before image acquisition. The PET acquisition was obtained in cranial-caudal direction, carried out from the external acoustic meatus to the root of the thigh; PET was reconstructed with a matrix of 128 × 128, ordered subset expectation maximum iterative reconstruction algorithm (two iterations, 28 subset), 8 mm Gaussian filter and 50 cm field of view. The CT acquisition parameters were the following: slice thickness 3.75 mm; 350 mA; 120 kV; tube rotation time 0.8 ms and collimation field of view (FOV) 50 cm. The CT images were reconstructed with a filtered back-projection. No iodate intravenous contrast was administered to patients.

### 2.3. ^18^F-FGD PET/CT Imaging Interpretation

^18^F-FDG PET/CT images were blindly reviewed by 2 nuclear physicians with more than 5 years of experience (C.F., N.M.) by using MultiVol PET/CT program (Volume Share 4.7 with Volume Viewer Software) of Advantage Workstation (GE Healthcare, Waukesha, Wisconsin, USA). Qualitative and semi-quantitative analysis were assessed both with and without the attenuation correction; non-attenuation corrected images were reviewed for final interpretation, in order to avoid artifacts induced by metallic components of CIED [21,22,23,24]. The visual qualitative analysis defined whether the ^18^F-FDG PET/CT was positive or negative for CIED infection. Positive ^18^F-FDG PET/CT was defined by increased ^18^F-FDG uptake around the device (generator pocket and/or leads) greater than mediastinal blood pool activity. Negative ^18^F-FDG PET/CT was defined if no increased ^18^F-FDG uptake around the device relative to surrounding tissues or mediastinal blood pool was detected [21,22,23]. Discordant analysis interpretation was discussed and resolved by consensus [23]. Semi-quantitative parameters were collected by volumes of interest (VOIs) semi-automatically drawn nearby the generator pocket and along the leads’ pathway. Maximum standardized uptake values (SUV_max_), normalized basing on the patient’s injected dose and weight, were collected from the VOI with the highest value. Semi-quantitative ratios (SQRs) were also calculated: SQR was defined as the maximum count rate in the region surrounding CIED over a mean count rate between normal left and right lung parenchyma; areas of abnormal lung parenchyma were avoided in the analysis.

### 2.4. Assessment of Patients’ Outcome

Management of patients and treatment decisions were made by the same cardiologists (D.C., C.D.A.) in all cases and established on the basis of the current clinical guidelines [12,25]. ^18^F-FDG PET/CT results were not used to guide the management decision. In patients who underwent surgery and whose CIED was removed, the final diagnosis was reached by the microbiological analysis. In the remnant patients, the final diagnosis was reached by clinical and instrumental follow-up, according to modified Duke’s criteria [25]. All patients were followed by the same cardiologists (D.C., C.D.A.) for 6 months after the performance of ^18^F-FDG PET/CT, both in case surgical CIED removal was performed and not.

### 2.5. Statistical Analysis

Sensitivity (Se), specificity (Sp), accuracy (Acc), positive predictive value (PPV) and negative predictive value (NPV) were calculated for CIEDIsusp patients. Reliability of ^18^F-FDG PET/CT qualitative analysis among the observers was evaluated with Cohen’s K. Semi-quantitative parameters were compared by Student’s *t*-test; *p* value lower than 0.05 was considered statistically significant. Ninety-five percent confidence intervals (95% CI) were added for all diagnostic accuracy parameters (Se, Sp, Acc, PPV, NPV). Receiver operating characteristic (ROC) curve analysis were performed to derive exploratory cut-off values. All statistical analysis was carried out using MedCalc^®^ Statistical Software version 2020 (MedCalc Software Ltd., Ostend, Belgium).

## 3. Results

### 3.1. Patients’ Baseline Characteristics

Demographic, clinical and instrumental characteristics about CIEDIsusp patients are reported in Table 1.

As regards the kinds of device implanted, 7/15 (47%) patients had implantable cardioverter defibrillators (ICDs) and 8/15 (53%) had pacemakers (PMs). Fever was present in 10/15 (67%) patients and local signs of pocket infection was found in 7/15 (47%) patients. Values of ESR and CRP were increased in 9/15 (60%) patients, while values of PCT and WBC were altered respectively in 4/15 (27%) and 1/15 (7%) patients. Blood cultures resulted positive in 3/15 (20%) patients and negative in the remnant 12/15 (80%). The bacteria identified were *Staphylococcus epidermidis* (n. 2) and *Staphylococcus aureus* (n. 1). TTE resulted positive in 1/15 (7%) patient (also confirmed by TEE); TEE was performed in 5/15 (33%) patients and it resulted positive in 4/5 patients; TEE was not performed in the remnants patients because of their clinical conditions.

The 15 ONCOctrl patients performed ^18^F-FDG PET/CT for oncological surveillance and they had no clinical suspicion of CIED infection. The devices implanted were ICDs in 5/15 (33%) and PMs in 10/15 (67%). They were affected by lung carcinoma (3/15), chronic lymphatic leukemia (2/15), non-Hodgkin lymphoma (2/15), kidney carcinoma (2/15), melanoma (3/15), intestinal carcinoma (2/15) and gastric cancer (1/15).

The mean time elapsing between the statement of possible CIED infection and the ^18^F-FDG PET/CT execution was 2 days (range: 1–3 days). During this time empirical antibiotic therapies were started: 4 patients assumed cefazolin, 2 amoxicillin + clavulanic acid, 3 amoxicillin + clavulanic acid and levoxacin, 1 daptomycin, 1 cefazolin and teicoplanin, one amoxicillin + clavulanic acid and daptomycin, 1 teicoplanin + ceftriaxone, 1 daptomycin and piperacillin tazobactam and 1 azithromycin and daptomycin and piperacillin tazobactam. The therapies did not interfere with the microbiological study of samples and did not invalidate ^18^F-FDG PET/CT results, because the short interval of time elapsing between the start of the therapy and the instrumental exam execution was not sufficient to obtain a complete bacterial count reduction so it did not influence ^18^F-FDG uptake.

The mean time elapsing between the CIED implantation and the ^18^F-FDG PET/CT execution was 3.2 years (range: 6 months–7 years) and the mean time elapsing between the ^18^F-FDG PET/CT execution and the surgical device removal was 5 days (range: 3–7 days). In the control group, the mean time elapsing between the CIED implantation and the ^18^F-FDG PET/CT execution was 3.5 years (range: 10 months–8 years).

### 3.2. ^18^F-FGD PET/CT Analysis Results

According to visual qualitative analysis, ^18^F-FDG PET/CT resulted positive in 11/15 (73%) CIEDIsusp patients; in the remnants 4/15 (27%) patients, the exam was considered negative [Figure 1, Figure 2].

Sites of infection were generator pocket in 9/11 and leads’ extracardiac pathway in the remnant 2/11 patients. No other pathological areas of ^18^F-FDG uptake were found even in the endocardium in any patient. All ONCOctrl patients did not show any abnormal ^18^F-FDG uptake area in proximity of the generator pocket or leads, so none of them was considered positive. The description of ^18^F-FDG PET/CT results is reported in Table 2.

### 3.3. Patients’ Outcome

The cardiac device was surgically removed in 12/15 (80%) CIEDIsusp patients; the entire pacing system was extracted intravenously. In these patients the final diagnosis was reached by the microbiological analysis: it was positive in 11/12 and negative in 1/12. In the remnant 3/15 (20%) patients who did not remove CIED the clinical and instrumental follow-up resulted negative for infection. The description of microbiological and clinical follow-up results is reported in Table 2. All ONCOctrl patients resulted negative during clinical follow-up.

### 3.4. Statistical Analysis Results

As regards the CIEDIsusp patients, 10/15 resulted as true positives (TPs), 3/15 true negatives (TNs) 1/15 false positive (FP) and 1/15 false negative (FN). As regards the ONCOctrl patients, agreement between ^18^F-FDG PET/CT and outcome was complete: all patients showed negativity in ^18^F-FDG PET/CT and resulted negative during clinical follow-up. In CIEDIsusp patients, Se, Sp, Acc, PPV and NPV of ^18^F-FDG PET/CT resulted 90.91% (95% CI: 58.72% to 99.77%), 75% (95% CI: 19.41% to 99.37%), 86.67% (95% CI: 59.54% to 98.34%), 90.91% (95% CI: 64.45% to 98.22%) and 75% (95% CI: 29.86% to 95.48%) respectively. Reproducibility among nuclear medicine physicians as regards qualitative analysis resulted as excellent (K value = 0.89).

In CIEDIsusp patients positive at ^18^F-FDG PET/CT, the mean value of SUV_max_ was 4.47 (range: 2.20–7.33; SD = 1.76) and the mean value of SQR was 7.15 (range: 2.88–11.63; SD = 3.11). In CIEDIsusp patients negative at ^18^F-FDG PET/CT, the mean value of SUV_max_ was 2.13 (range: 1.82–2.56; SD = 0.32) and the mean value of SQR was 2.82 (range: 1.43–3.37; SD = 0.93). In ONCOctrl patients, the mean value of SUV_max_ was 1.98 (range: 1.29–2.96; SD = 0.50) and the mean value of SQR was 3.48 (range: 1.90–5.38; SD = 0.93) [Figure 3].

In patients with diagnosis of CIED infection, the mean value of SUV_max_ was 3.91 (range: 1.96–7.33; SD = 1.87). In patients without CIED infection, the mean value of SUV_max_ was 2.14 (range: 1.29–4.96; SD = 0.86). The difference between them was statistically significant (*t* = 3.35; 95% CI: 0.68% to 2.85%; *p* < 0.05) [Figure 4a]. The mean value of SQR was 6.07 (range: 2.88–11.63; SD = 3.17) in patients positive for CIED infection and 3.72 (range: 1.43–9.54; SD = 1.8) in negative ones. The difference was statistically significant (*t* = 2.57; 95% CI: 0.40% to 4.29%; *p* < 0.05) [Figure 4b].

Exploratory SUV_max_ cut-off value resulted as 2.56 (area under the curve (AUC) = 0.957; ES = 0.032; 95% CI: 0.395% to 0.519%) while exploratory SQR cut-off value resulted as 4.15 (AUC = 0.878; ES = 0.071; 95% CI: 0.239% to 0.517%).

## 4. Discussion

Our study aims to assess the value of ^18^F-FDG PET/CT in patients who referred to cardiologist for suspected CIED infection. Previous studies already evaluated the role of ^18^F-FDG PET/CT in these patients, but there was not homogeneity in methodological criteria [22,24,26,27,28]. In our study all patients with suspicion of CIED infection were instructed to follow a high-fat/low-carbohydrate (HFLC) diet that allowed an optimal suppression of physiological myocardial glucose utilization, facilitating the evaluation of intracardiac sites of elevated ^18^F-FDG uptake. A correct diet before ^18^F-FDG PET/CT was useful to reduce the physiological myocardial uptake avoiding either false-positive (physiological uptake interpreted as infection) or false-negative results (infectious uptake unrecognized because of predominant diffuse physiological uptake) [21,25,29,30]. Although a precise protocol for ^18^F-FDG PET/CT is not completely standardized for cardiac infection imaging, it is highly recommended a dietary preparation with 1 or 2 meals of high fat and low carbohydrates followed by a fasting period of at 8 h [14].

Furthermore, in all patients with suspicion of CIED infection the timing of ^18^F-FDG PET/CT was accurately defined in order not to be influenced by antibiotic therapy. According to the current guidelines, empirical antimicrobial therapies have to be commenced as soon as possible in patients with suspected CIED infection [12] and in our study, ^18^F-FDG PET/CT was performed no more than 2 days after the beginning of them; this period of time was not sufficient to interfere with ^18^F-FDG uptake. Previous studies with the same aim as ours, even if analyzed in a larger population, presented a bias of the lack of dietary preparation and duration of antibiotic therapy that resulted from being performed for weeks before ^18^F-FDG PET/CT, influencing the results [26,27,28].

In our study all patients were evaluated by the same cardiologist and ^18^F-FDG PET/CT results were not used to guide the management of patients; treatment decisions were established on the basis of the current guidelines [12,25,31]. In the majority of patients (12/15) CIED was removed and the microbiological analysis was performed while in the remnants (3/15), for whom cardiologist did not remove the device, patients’ outcome was assessed by follow-up according to modified Duke’s criteria [25,32].

About patients with CIED infection, even if ^18^F-FDG PET/CT is not recommended for routine performance, it is mentioned as a potential useful additive tool in selected cases, in particular when there is uncertainty about generator pocket infection [14,20,25,26,33].

Our study revealed a good reliability of ^18^F-FDG PET/CT thanks to the good agreement with final outcomes. Discordance was observed only in two patients. In one patient ^18^F-FDG PET/CT was considered positive for CIED infection, but the clinical follow-up resulted in being negative; this false positive result has been ascribed to the presence of a foreign-body inflammation reaction nearby the pocket. In this patient the suspicion of CIED infection was postulated on the basis of the presence of fever and increased values of inflammatory indexes, without local signs of generator pocket infection. The cardiologist decided not to surgically remove the device, because the patient was in a good general health state and both fever and inflammatory indexes were slightly increased. The patient underwent close clinical monitoring, showing rapid temperature decrease and laboratory indexes normalization. These findings supported the hypothesis of high ^18^F-FDG PET/CT uptake due to tissue’s inflammatory reaction as also reported in literature [21]. In one patient ^18^F-FDG PET/CT resulted negative, but the microbiological analysis after CIED removal showed *Staphylococcus epidermidis* infection; the false negative result can be explained by the small site of the infection near the electrocatheter, less than the resolution of the technique [13,26,27,28,34].

Our results from the diagnostic performance of ^18^F-FDG PET/CT were also encouraging, revealing better results for sensitivity and PPV (both 90.91%) in line with those reported in literature (80–97%) [21,23,35,36]. In our analysis these good results, such as 86.67% accuracy, can be at least in part attributed to the high number of pocket infection of our patients’ cohort. The specificity and NPV resulted lower (both 75%) because of the ^18^F-FDG uptake also in unspecific inflammatory conditions; other factors that can influence the cardiac glucose uptake of ^18^F-FDG are sugar blood level, insulin blood level, left vs right ventricle blood shunt, vasculitis and ateromatous arteries [4,7].

In our study the reliability of visual qualitative analysis of ^18^F-FDG PET/CT images by the 2 nuclear medicine physicians resulted in being excellent (K = 0.89). It was thanks to the high quality of images both corrected and non-corrected for attenuation that artifacts related to the metallic components of the device were avoided [22,23]. Our results confirmed those of Granados et al. and Bensimhon et al.; in their studies, K values for presence of CIED infection were, respectively, 0.81 and 0.80 [22,36].

In addition to visual qualitative assessment, semi-quantitative parameters were also collected in order to investigate their usefulness in the diagnosis of CIED infection. In literature there is no consensus about the choice of them and their evaluation; we have chosen the SUV_max_ because it is the most validated parameter and SQR because it allows any error in the radiotracer uptake detection during the attenuation correction process to be avoided [21,22,23,24].

In our study, the differences of mean values of SUV_max_ and SQR between patients with confirmed and unconfirmed CIED infection were statistically significant (*p* < 0.05) suggesting that these values could further contribute to the correct interpretation of ^18^F-FDG PET/CT images; however, it is opportune to underline that these results must be interpreted with caution due to the small size of our sample and the overlap of the ranges of values collected.

Besides these limitations, it was possible to propose exploratory cut-off values of SUV_max_ and SQR, which are useful to discriminate patients with CIED infection from negative ones. In our study SUV_max_ cut-off value resulted 2.56 and SQR cut-off value 4.15, obtained by ROC analysis. Also, Bensimhon et al. proposed a SUV_max_ cut-off value of 2.2; Ahmed et al. and Sarrazin et al. proposed SQR cut-off values of 2.00 and 1.87 respectively [21,23,36]. Our results are similar to those reported in literature about SUV_max_ and different about SQR, but it is important to consider that differences in methodological statistical analysis and in the population samples analyzed can interfere with the comparison.

^18^F-FDG PET/CT is whole-body multimodality imaging; its most relevant advantages are the evaluation of extracardiac components of the device, which are beyond the echocardiographic field of view, the detection of unexpected sources of primary infection, and the identification of embolic consequences of endocarditis [14,23,35,37]. In our study no sites of infection different from those of CIED were detected, excluding more complicated diseases. Otherwise ^18^F-FDG PET/CT uses ionizing radiation, but current technologies reduce considerably the radiation exposure; it should be also remembered that CIED patients are generally of high age and for them the risks of CIED infection are more serious than those potentially of ionizing radiation [14].

Although our study showed promising results, it is not devoid of limitations such as the retrospective and monocentric nature of our analysis; our sample was small, but homogeneous and in line with samples reported in literature; conventional instrumental exams such as TEE and CIED removal were not always performed, but this reflects the real situations that clinicians have to face.

## 5. Conclusions

This study suggests the clinical usefulness of ^18^F-FDG PET/CT in patients with CIED infection due to its high sensitivity, repeatability and non-invasiveness. It can help the clinicians in decision making, especially in patients with doubtful clinical presentation, and it should be considered as a possible methodological step into the flowchart of management of patients with suspected CIED infection. Future large-scale and multicentric studies should be conducted to establish precise protocols about ^18^F-FDG PET/CT performance.

## Figures and Tables

**Figure 1 jcm-09-02246-f001:**
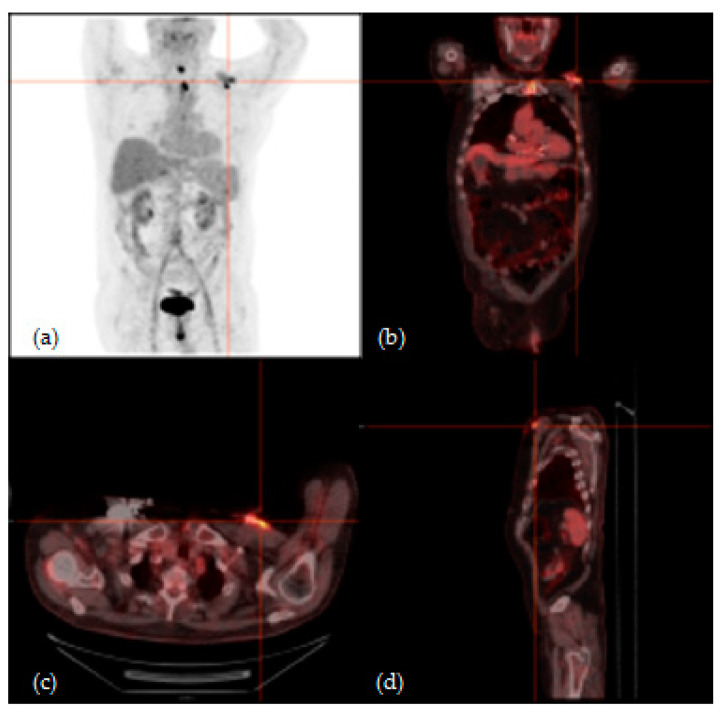
^18^F-fluorodeoxyglucose positron-emission tomography/computed tomography (^18^F-FDG PET/CT) of a 78-year-old man with suspicion of cardiac implantable electronic device (CIED) infection because of bacterial blood culture positive for *Staphylococcus epidermidis*. (**a**) Maximum intensity projection (MIP), (**b**) coronal fusion, (**c**) axial fusion and (**d**) sagittal fusion images showed increased ^18^F-FDG uptake involving the CIED pocket. After surgical CIED removal, microbiological analysis of explanted materials confirmed *Staphylococcus epidermidis* infection.

**Figure 2 jcm-09-02246-f002:**
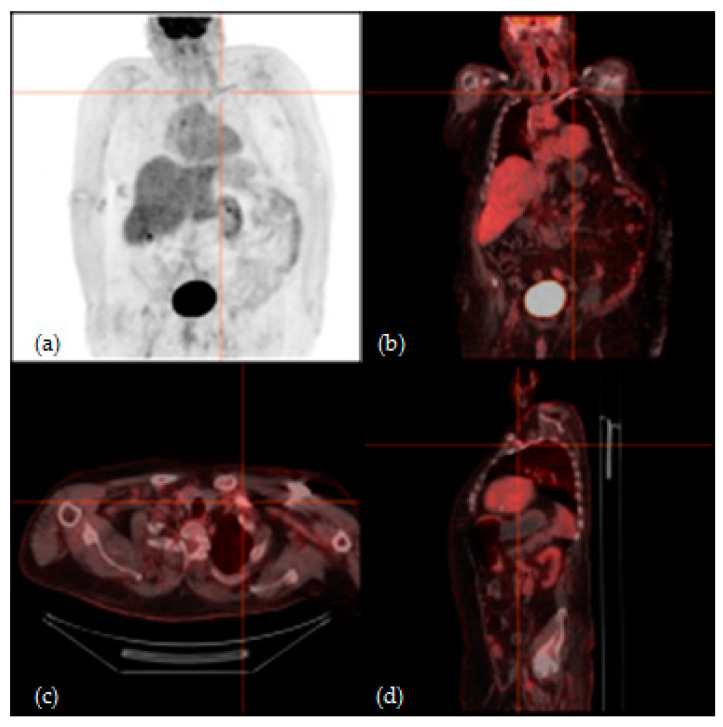
^18^F-FDG PET/CT of a 56-year-old man with suspicion of CIED infection; both TTE and TEE were positive, but bacterial blood culture was negative. (**a**) Maximum intensity projection (MIP), (**b**) coronal fusion, (**c**) axial fusion and (**d**) sagittal fusion images showed increased ^18^F-FDG uptake involving the leads. After surgical CIED removal, microbiological analysis of explanted materials showed *Staphylococcus epidermidis* infection.

**Figure 3 jcm-09-02246-f003:**
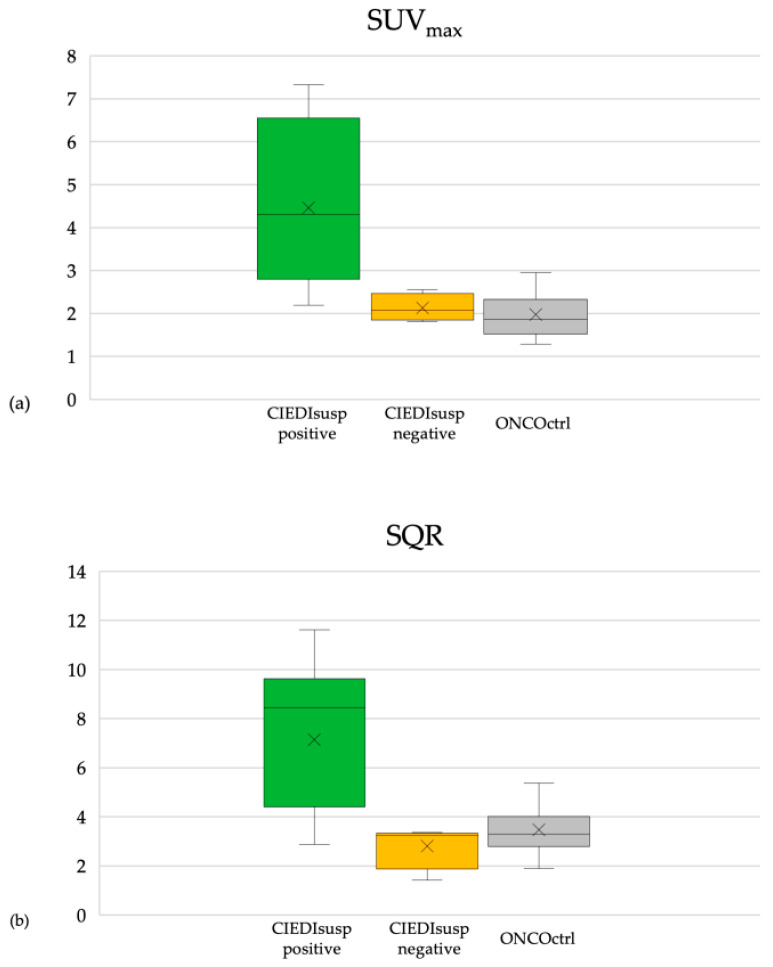
(**a**) SUV_max_ distribution in CIEDIsusp patients positive at ^18^F-FDG PET/CT, CIEDIsusp patients negative at ^18^F-FDG PET/CT and oncological surveillance without clinical suspicion of CIED infection group (ONCOctrl) patients. (**b**) SQR distribution in CIEDIsusp patients positive at ^18^F-FDG PET/CT, CIEDIsusp patients negative at ^18^F-FDG PET/CT and ONCOctrl patients.

**Figure 4 jcm-09-02246-f004:**
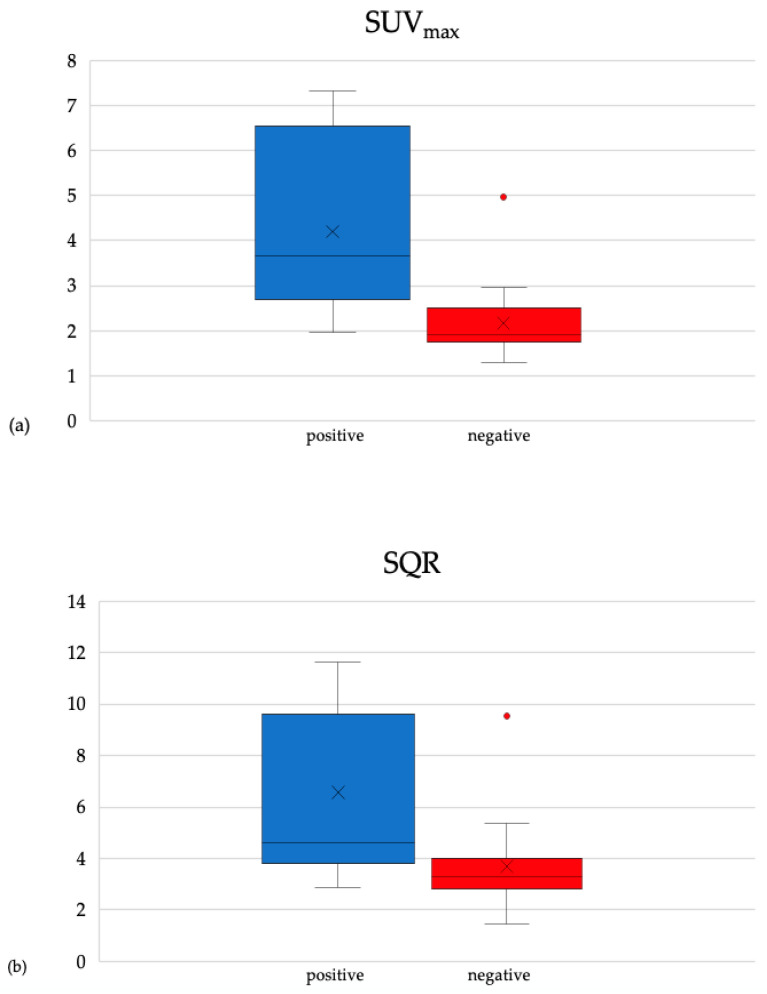
(**a**) SUV_max_ distribution in patients with CIED infection (positive) and patients without CIED infection (negative). (**b**) SQR distribution in patients with CIED infection (positive) and patients without CIED infection (negative).

**Table 1 jcm-09-02246-t001:** Demographic, clinical and instrumental characteristics of cardiac implantable electronic devices infection group (CIEDIsusp) patients.

Patients				Clinical Signs		Laboratory Signs					Instrumental Signs	
ID	Age	Sex	Type of CIED Implanted	Fever	Local Signs of CIED Infection	ESR (v.n. <20 mm/h)	CRP (v.n. <2.9 mg/L)	PCT (v.n. <0.5 ng/mL)	WBC (v.n. 3.7-9.7 x10^3^ uL)	Blood Culture	TTE	TEE
1	75	M	ICD	Yes	Yes	48	18.4	16	5.6	0	Negative	/
2	74	M	PM	Yes	No	8	3.1	3.7	5.5	0	Negative	/
3	73	M	PM	Yes	No	18	35	0.02	10.05	0	Negative	/
4	83	M	PM	Yes	Yes	25	1	0.03	7.65	*Staphyl. Aureus*	Negative	Negative
5	83	M	ICD	Yes	Yes	48	3	0.04	6.57	*Staphyl. Epiderm.*	Negative	/
6	59	F	PM	No	Yes	38	0	0.02	6.7	0	Negative	/
7	46	M	PM	Yes	Yes	52	42.7	0.07	7.77	0	Negative	Positive
8	84	M	ICD	Yes	No	50	70.6	11	6.2	0	Negative	/
9	56	M	PM	No	No	31	99	0.13	8.91	0	Positive	Positive
10	63	M	PM	Yes	No	50	3.1	3.4	4.35	0	Negative	/
11	71	M	PM	Yes	Yes	0	1	0.02	5.48	0	Negative	Positive
12	53	M	ICD	No	No	16	1.7	0.03	6.31	0	Negative	/
13	73	M	ICD	Yes	No	37	38	0.07	7.04	0	Negative	Positive
14	62	M	ICD	No	Yes	0	0	0	4.5	0	Negative	/
15	78	M	ICD	No	No	1	1	0.03	5.93	*Staphyl. Epiderm.*	Negative	/

CIED: cardiac implantable electronic device; ESR: erythrosedimentation rate; CRP: C-reactive protein; PCT: procalcitonin; WBC: white blood cells; TTE: trans-thoracic echocardiography; TEE: trans-esophageal echocardiography; ICD: implantable cardioverter defibrillators; PM: pacemakers.

**Table 2 jcm-09-02246-t002:** ^18^F-FDG PET/CT and final results in CIEDIsusp patients.

18F-FDG PET/CT Analysis Results						Final Results	
ID	Result	Site	SUVmax	SQR	Microbiological Analysis		Clinical Follow-Up
1	Positive	Pocket	3.65	3.80	*Staphyl. haemolyticus*		/
2	Positive	Pocket	4.96	9.54	/		Negative
3	Negative	Pocket	2.19	3.37	Negative		/
4	Positive	Pocket	3.32	4.61	*Staphyl. aureus*		/
5	Positive	Pocket	4.31	8.71	*Staphyl. epidermidis*		/
6	Negative	Pocket	2.56	3.24	/		Negative
7	Positive	Lead	2.80	4.41	*Staphyl. epidermidis*		/
8	Negative	Pocket	1.96	3.24	*Staphyl. epidermidis*		/
9	Positive	Lead	2.69	2.88	*Staphyl. epidermidis*		/
10	Positive	Pocket	6.55	9.63	*Staphyl. epidermidis*		/
11	Positive	Pocket	6.75	10.38	*Staphyl. epidermidis*		/
12	Negative	Pocket	1.82	1.43	/		Negative
13	Positive	Pocket	2.20	4.58	*Staphyl. epidermidis*		/
14	Positive	Pocket	4.61	8.46	*Staphyl. epidermidis*		/
15	Positive	Pocket	7.33	11.63	*Staphyl. epidermidis*		/

SUV_max_: maximum standardized uptake values; SQR: semi-quantitative ratio.

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
