# Peer review of "Usefulness of 18F-FDG PET/CT in Patients with Cardiac Implantable Electronic Device Suspected of Late Infection"

_jcm, 2020, doi:10.3390/jcm9072246_

Round 1
Reviewer 1 Report
Colleagues Rubini et al. must be complimented on their clearly and transparently written paper. I have, though, three general comments and numerous specific comments which I reckon will help improving the paper further. Keep up the good work.
General comments:
(1) I propose to add group names to the two groups of 15 patients each, for instance, CIEDsusp and ONCO group. That would increase transparency and help the reader instantaneously follow along your lines of argumentation.
(2) The authors have been quite generous in adding line breaks (often after one to two sentences only). Please combine all paragraphs to one which logically belong together, thanks.
(3) The discussion section is written as a prose and can definitely be streamlined by adding subsections to it, thereby following more clearly a thread of argumentation and avoiding redundant replications of parts of the results section. Please restructure the discussion, for instance according to the Docherty & Smith editorial from 1999; The case for structuring the discussion of scientific papers. BMJ. 1999;318(7193):1224‐1225. https://doi.org/10.1136/bmj.318.7193.1224
Specific comments:
l.26 in patients
l.34 ”Exploratory, data-driven” instead of “Hypothetical”
l.55-56: “potentially accompanied by” instead of “not avoided by”
l.57: propose “help clinicians choosing the most appropriate treatment”
l.63: more recent references are the paper “An update on the unparalleled impact of FDG-PET imaging on the day-to-day practice of medicine with emphasis on management of infectious/inflammatory disorders” by Alavi et al. (https://doi.org/10.1007/s00259-019-04490-6) and the Special Issue “FDG-PET/CT Imaging in Infectious and Inflammatory Disorders” in PET Clinics, April 2020 (editorial: https://doi.org/10.1016/j.cpet.2020.01.002). Reference #18 dates back to 2011.
l.42-63 should be one paragraph; please delete line breaks in between. Please revisit also the rest of the text (see general comment (2) above).
l.66 Materials and Methods
l.70 How was written informed consent gathered – were these patients invited for follow-up visits? During which time frame (month/year to month/year) were FDG-PET/CT scans for the sake of this retrospective study performed?
l.113 two nuclear physicians: add initials if these also are authors of the manuscript
l.133 same (“by the same cardiologist”)
l.141-142 Why Cohen’s kappa? Is this a question of reliability (de Vet et al.: When to use agreement versus reliability measures; https://doi.org/10.1016/j.jclinepi.2005.10.015)? According to GRRAS (Kottner et al.: Guidelines for Reporting Reliability and Agreement Studies (GRRAS) were proposed; https://doi.org/10.1016/j.ijnurstu.2011.01.016), proportions of agreement come to mind; however, these coincide with the accuracy you already reported. Please clarify the aim of using Cohen’s kappa here; the term “concordance” does actually remind on of concordance correlation coefficients for continuous measures.
l.145-146 Same as above.
l.149-150 replace “individuate hypothetical” by “derive exploratory”
l.150: Please add “Ninety-five percent confidence intervals (95% CI) were added for all diagnostic accuracy parameters (Se, Sp and so on).” 95% CI for AUC-ROC are actually already included; those for Se, Sp, Acc, PPV, NPV need to be added (see comment below to the Results section).
l.151 Please add other software solutions (like Medcalc) if needed.
l.155 Table [not table]; same in l.211,220
l.170 Group reference will be much more accessible with group names according to general comment (1). Another example: l.236. With group titles, you don’t need to describe the respective groups every time.
l.223-224 & l.234-235 See comments to statistical analysis above. What are the assessments based upon? Landis & Koch, The Measurement of Observer Agreement for Categorical Data, Biometrics 1977;33(1),159-174?
l.225-228 I would clearly prefer concrete numbers for true positive (n=10), true negative (n=3), false positive (n=1), and false negative (n=1). This can easily be achieved by adding TP, TN, FP, FN, for instance, as follows: “…in 10/15 patients (true positives)”.
l.233 95% CIs need to be added! This can, for instance, easily be achieved by MEDCALC’s diagnostic test evaluator here: www.medcalc.org/calc/diagnostic_test.php (if EXCEL should fail you providing exact (Clopper-Pearson type) 95% CI for binomial proportions). Just insert a=10, b=c=1, d=3 & apply “Test”.
l.238-239 Please add an estimate for the mean difference and a respective 95% CI. That quantifies the magnitude of the difference – in opposition to mere p-values.
l.244-245 Same.
l.249 Replace “Hypothetical” by “Exploratory, data-driven”
l.364 Please add page numbers and DOI.
l.444 Update reference which was published in 2015.
Author Response
General comments
- I propose to add group names to the two groups of 15 patients each, for instance, CIEDsusp and ONCO group. That would increase transparency and help the reader instantaneously follow along your lines of argumentation.
Thank you for this suggestion; we renamed the two groups “CIEDIsusp” and “ONCOctrl” respectively (new version page 2, lines 71, 81). In the main text, further changes have been applied (new version page 2, line 83; page 3, line 129; page 4, lines 138, 140, 153; page 5, line 174; page 6, lines 190, 193; page 7, lines 196, 201, 203-204, 206).
- The authors have been quite generous in adding line breaks (often after one to two sentences only). Please combine all paragraphs to one which logically belong together, thanks.
We surely agree with this comment, so all the manuscript has been revised combining paragraphs that belong together.
- The discussion section is written as a prose and can definitely be streamlined by adding subsections to it, thereby following more clearly a thread of argumentation and avoiding redundant replications of parts of the results section. Please restructure the discussion, for instance according to the Docherty & Smith editorial from 1999; The case for structuring the discussion of scientific papers. BMJ. 1999;318(7193):1224‐1225. https://doi.org/10.1136/bmj.318.7193.1224.
Thank you for signaling this interesting editorial; following its suggestions, we recombined some paragraphs of the discussion and removed the initial statements because the redundancy you rightly observed.
Specific comments
- l.26 in patients
The correction has been applied (new version page 1, line 26).
- l.34 ”Exploratory, data-driven” instead of “Hypothetical”
Thank you for this suggestion, the word “hypothetical” has been replaced by “exploratory” each time it appeared in the main text (new version page 1, line 33; page 3, line 134; page 8, lines 224-225; page 9, line 293).
- l.55-56: “potentially accompanied by” instead of “not avoided by”
We are glad of this improvement; it has been applied in new version of the manuscript at page 2, line 53.
- l.57: propose “help clinicians choosing the most appropriate treatment”
We also like this suggestion; it has been applied in new version of the manuscript at page 2, line 55.
- l.63: more recent references are the paper “An update on the unparalleled impact of FDG-PET imaging on the day-to-day practice of medicine with emphasis on management of infectious/inflammatory disorders” by Alavi et al. (https://doi.org/10.1007/s00259-019-04490-6) and the Special Issue “FDG-PET/CT Imaging in Infectious and Inflammatory Disorders” in PET Clinics, April 2020 (editorial: https://doi.org/10.1016/j.cpet.2020.01.002). Reference #18 dates back to 2011.
Thank you for the suggestion; we removed the reference #18 dated back to 2011 and updated bibliography as suggested with new references #18 and #19 (new version page 2, line 60; page 11, lines 387-393).
- l.42-63 should be one paragraph; please delete line breaks in between. Please revisit also the rest of the text (see general comment (2) above).
As already answered comment 2, all the manuscript, including this part, has been revised combining paragraphs that belong together.
- l.66 Materials and Methods
We entitled the paragraph “Experimental Section” following the template of JCM; anyway the correction suggested has been applied in new version of the manuscript at page 2, line 63.
- l.70 How was written informed consent gathered – were these patients invited for follow-up visits? During which time frame (month/year to month/year) were FDG-PET/CT scans for the sake of this retrospective study performed?
Following the reviewer’s suggestion, the time frame during which 18F-FDG PET/CT scans were performed has been inserted in the new version of the manuscript at page 2, line 68. Furthermore explanation about informed consent and follow-up are reported at page 2, line 67 and page 3, lines 119-126 respectively.
- l.113 two nuclear physicians: add initials if these also are authors of the manuscript
Thank you for this suggestion, we added initials of the two nuclear physicians who reviewed 18F-FDG PET/CT images (new version page 3, line 101).
- l.133 same (“by the same cardiologist”)
We did this modification same as above (new version page 3, lines 119-120, 125).
- l.141-142 Why Cohen’s kappa? Is this a question of reliability (de Vet et al.: When to use agreement versus reliability measures; https://doi.org/10.1016/j.jclinepi.2005.10.015)? According to GRRAS (Kottner et al.: Guidelines for Reporting Reliability and Agreement Studies (GRRAS) were proposed; https://doi.org/10.1016/j.ijnurstu.2011.01.016), proportions of agreement come to mind; however, these coincide with the accuracy you already reported. Please clarify the aim of using Cohen’s kappa here; the term “concordance” does actually remind on of concordance correlation coefficients for continuous measures.
We amended this point; as the reviewer correctly stated, agreement between 18F-FDG PET/CT findings and patients’ final outcome is represented by Accuracy, already reported in the manuscript (new version page 7, line 208).
- l.145-146 Same as above.
We carefully read the articles suggested by the reviewer and they surely advise against using Cohen’s K in the comparison between diagnostic tool and final diagnosis. Conversely, according to its definition, it can be applied for the evaluation of the agreement between ratings made by two different clinicians (interrater reliability). For this reason, we deemed appropriate to keep this evaluation and we better explained it by introducing the term reliability (new version page 3, line 129; page 9, line 278).
- l.149-150 replace “individuate hypothetical” by “derive exploratory”
As mentioned above, the word “hypothetical” has been replaced by “exploratory” each time it appeared in the main text (new version page 1, line 33; page 3, line 134; page 8, lines 224-225; page 9, line 293).
- l.150: Please add “Ninety-five percent confidence intervals (95% CI) were added for all diagnostic accuracy parameters (Se, Sp and so on).” 95% CI for AUC-ROC are actually already included; those for Se, Sp, Acc, PPV, NPV need to be added (see comment below to the Results section).
- l.151 Please add other software solutions (like Medcalc) if needed.
As the reviewer correctly suggested, MedCalc is a more suitable software for medical statistical analysis; so we repeated all the statistical tests by using it (new version page 3, lines 135). Furthermore the sentence suggested and values of 95% CI have been introduced as requested (new version page 3, lines 132-133; page 7, lines 207-209, 213-214, 220).
- l.155 Table [not table]; same in l.211,220
Thank you noting it; the corrections have been applied (new version page 4, line 139; page 6, line 192; page 7, line 200).
- l.170 Group reference will be much more accessible with group names according to general comment (1). Another example: l.236. With group titles, you don’t need to describe the respective groups every time.
As suggested in the first general comment, group names have been added and the adjustment has been applied in all the main text (new version page 2, lines 71, 81, 83; page 3, line 129; page 4, lines 138, 140, 153; page 5, line 174; page 6, lines 190, 193; page 7, lines 196, 201, 203-204, 206).
- l.223-224 & l.234-235 See comments to statistical analysis above. What are the assessments based upon? Landis & Koch, The Measurement of Observer Agreement for Categorical Data, Biometrics 1977;33(1),159-174?
As we said above, any reference to Cohen’s K in the context of diagnostic accuracy was removed.
- l.225-228 I would clearly prefer concrete numbers for true positive (n=10), true negative (n=3), false positive (n=1), and false negative (n=1). This can easily be achieved by adding TP, TN, FP, FN, for instance, as follows: “…in 10/15 patients (true positives)”.
Thank you for this suggestion; the correction has been applied (new version page 7, lines 203-204).
- l.233 95% CIs need to be added! This can, for instance, easily be achieved by MEDCALC’s diagnostic test evaluator here: www.medcalc.org/calc/diagnostic_test.php (if EXCEL should fail you providing exact (Clopper-Pearson type) 95% CI for binomial proportions). Just insert a=10, b=c=1, d=3 & apply “Test”.
- l.238-239 Please add an estimate for the mean difference and a respective 95% CI. That quantifies the magnitude of the difference – in opposition to mere p-values.
- l.244-245 Same.
These valuable suggestions have been considered; all the tests have been repeated as mentioned above and 95% CI have been added (new version page 7, lines 207-209, 213-214, 220).
- l.249 Replace “Hypothetical” by “Exploratory, data-driven”
As suggested above, the word “hypothetical” has been replaced by “exploratory” each time it appeared in the main text (new version page 1, line 33; page 3, line 134; page 8, lines 224-225; page 9, line 293).
- l.364 Please add page numbers and DOI.
Thank you for this suggestion; the correction has been applied (new version page 10, lines 334-335).
- l.444 Update reference which was published in 2015.
Thank you for this suggestion; the correction has been applied (new version page 12, line 419).
Reviewer 2 Report
LS,
I have read with interest the paper by Rubini et al. describing their observational retrospective analysis of 30 FDG PET/CT scans in patients with a CIED, 15 for suspicion of infection and 15 for oncology. Final diagnosis was based on microbiology of explanted materials in the majority of the former group (12/15) and clinical and laboratory follow-up in the remainder.
The authors report impressive values for sensitivity and PPV especially, but still good values for accuracy, specificity and NPV as well.
There are a number of findings in this cohort that I wish to highlight (also in comparison to other comparable publications):
Firstly, the incidence of endocarditis is high (11/15 ≈ 73%) compared to some other papers (Bensimhon et al. 2011: 10/21 – 48%; Tlili et al. 2015: 18/40 – 45%) but comparable to others (Leccisotti et al. 2014: 22/27 – 81.5%).
Secondly, the number of patients with pocket-site infection was high compared to lead(-only) infection, 9/11 vs. 2/11.
Taking this into account, I have a number of questions to ask and suggestions to make to the authors of the current paper:
- Study design: The study design was observational retrospective, yet the authors claim that findings on FDG PET/CT was not taken into account when deciding on the management (Lines 29-30: All patients underwent standard clinical management regardless 18F-FDG PET/CT results and lines 133-135: Management of patients and treatment decisions were made by the same cardiologist in all cases and established on the basis of the current clinical guidelines [12,24]. 18F-FDG PET/CT results were not used to guide the management decision.)
If the study was retrospective in design, why were FDG PET/CT scans performed if they did not inform the management decisions? If performed in the clinical routine, I assume they were made to help in diagnosis. If the scans were performed for research purposes, the design was not retrospective.
- Statistical analysis: Lines 141-142: Concordance between 18F-FDG PET/CT findings and patients’ outcome was evaluated by Cohen’s K. This does not make sense to me. I am not a statistician by training, but kappa is used to take into account the possibility that two readers agree purely based on chance. That assumption does not make much sense in the comparison between diagnostic tool and final diagnosis. In this case, Accuracy is adequate in explaining the concordance.
- Methods: In the image text, it is stated that “After surgical CIED removal, bacterial blood culture showed Staphylococcus Epidermidis infection”. I assume the authors mean culture of the explanted materials?
- Analysis/figs 3&4: Sens, Spec, Acc, PPV and NPV are calculated based on the 15 patients with suspicion of infection. I agree with this. However, in the analysis after that (line 236 and onward) it appears the negative (oncology) controls have been added to the analysis and conflated with the 3 scans with a final diagnosis of no infection. This should be explained better. Additionally, by adding the oncological scans, the false positive scan with a high SUVmax and SQR appears to be treated as an outlier. This is one of the most important findings in the study, which has been swept away rather unceremoniously.
I would prefer the figures to show three clusters: CIED infection (positive) [11/15], CIED infection (negative) [4/15] and Oncology (no infection) [15/15] to better show the values for each of these groups. I also feel the SUV and SQR values should be reported for each of these groups.
- Discussion: Please explain “this false positive result has been ascribed to the presence of a foreign-body inflammation reaction nearby the pocket” in more detail. Is this based on cytology or biopsy? Were there clinical findings to support this? Is this purely an assumption without further available proof? If so, what publications support your assumption?
- Discussion: and likewise for “the false negative result can be explained by the small site of the infection near the elettrocatheter, less than the resolution of the technique”: Was this proven by focus detection after extraction? Merely an assumption?
- Discussion: line 306: “left vs right ventricle” – I don’t understand what you mean by this, please elaborate.
- Discussion: If a direct comparison in SUVmax and SQR is not possible between studies, what is the value of reporting them? Please elaborate further.
- Discussion: It should be explained that FDG PET/CT is more accurate in pocket infection than it is in lead infection, and that the high levels of sensitivity found are probably due (at least in part) to the high number of pocket infections in this cohort.
- Conclusions: “It can help the clinicians in decision making, (…)” – This leads me back to question 1: How can you conclude FDG PET/CT can help in decision making when your study design explicitly did not let the PET/CT results influence clinical decision making?
Thank you for allowing me to review your work.
Author Response
- The Study design: The study design was observational retrospective, yet the authors claim that findings on FDG PET/CT was not taken into account when deciding on the management (Lines 29-30: All patients underwent standard clinical management regardless 18F-FDG PET/CT results and lines 133-135: Management of patients and treatment decisions were made by the same cardiologist in all cases and established on the basis of the current clinical guidelines [12,24]. 18F-FDG PET/CT results were not used to guide the management decision.)
If the study was retrospective in design, why were FDG PET/CT scans performed if they did not inform the management decisions? If performed in the clinical routine, I assume they were made to help in diagnosis. If the scans were performed for research purposes, the design was not retrospective.
The use of 18F-FDG PET/CT in cases of infections and inflammations has been widely proven even with location-related differences. 18F-FDG PET/CT in patients who have generic signs of infection such as fever and/or elevation of inflammatory indexes can be useful to identify any infectious focus throughout the body, including the sites of CIEDs. This allowed us to carry out this investigation in clinical practice, in the times and in ways also useful for the evaluation of the endocardium. Just starting from this premise, we frame our study as a retrospective observational one. Furthermore oncology patients with CIED were also recruited from those who performed the routine cardiological checks by the cardiologists involved in the study (D.C. and C.D.A.). On the other hand, 18F-FDG PET/CT for the diagnosis of endocarditis is currently not provided in common clinical practice and for this reason cardiologists made their decisions independently from PET/CT. Our work wants to fit right into this clinical scenario, and provide further results in support or not of the usefulness of PET/CT in this field.
- Statistical analysis: Lines 141-142: Concordance between 18F-FDG PET/CT findings and patients’ outcome was evaluated by Cohen’s K. This does not make sense to me. I am not a statistician by training, but kappa is used to take into account the possibility that two readers agree purely based on chance. That assumption does not make much sense in the comparison between diagnostic tool and final diagnosis. In this case, Accuracy is adequate in explaining the concordance.
We amended this point; as the reviewer correctly stated, agreement between 18F-FDG PET/CT findings and patients’ final outcome is represented by Accuracy, already reported in the manuscript (new version page 7, line 208).
- Methods: In the image text, it is stated that “After surgical CIED removal, bacterial blood culture showed Staphylococcus Epidermidis infection”. I assume the authors mean culture of the explanted materials?
Thank you for noting this mistake; as the reviewer correctly assumed, the culture is referred to the explanted materials. We corrected the sentence in the new version of the manuscript at page 5, lines 180-181 and page 6, lines 186-187.
- Analysis/figs 3&4: Sens, Spec, Acc, PPV and NPV are calculated based on the 15 patients with suspicion of infection. I agree with this. However, in the analysis after that (line 236 and onward) it appears the negative (oncology) controls have been added to the analysis and conflated with the 3 scans with a final diagnosis of no infection. This should be explained better. Additionally, by adding the oncological scans, the false positive scan with a high SUVmax and SQR appears to be treated as an outlier. This is one of the most important findings in the study, which has been swept away rather unceremoniously.
I would prefer the figures to show three clusters: CIED infection (positive) [11/15], CIED infection (negative) [4/15] and Oncology (no infection) [15/15] to better show the values for each of these groups. I also feel the SUV and SQR values should be reported for each of these groups.
We have carefully considered these comments, which we believe are closely related to each other, and they have been a reason for further meditation for us. The main goal of 18F-FDG PET/CT is correctly identify patients classified by outcome in positive or negative and semiquantitative parameters’ collection is a following step. We included the entire population because it is sensible to compare mean values of SUVmax and SQR between positive and negative. In addition, this evaluation and representation highlights what the reviewer rightly points out to be one of the most important results, that is the false positive patient, which was better and more effectively described in discussion as requested (new version page 9, lines 261-267). The suggestion to describe the 3 above mentioned clusters can certainly add data which, however, would prevent us from collecting diagnostic measures and distract from the most relevant results; however, if the reviewer considers it necessary, we are available to add what is requested.
- Discussion: Please explain “this false positive result has been ascribed to the presence of a foreign-body inflammation reaction nearby the pocket” in more detail. Is this based on cytology or biopsy? Were there clinical findings to support this? Is this purely an assumption without further available proof? If so, what publications support your assumption?
We are glad of this comment, so a clearer explanation of the false positivity has been added in the main text, as mentioned above (new version page 9, lines 261-267).
- Discussion: and likewise for “the false negative result can be explained by the small site of the infection near the elettrocatheter, less than the resolution of the technique”: Was this proven by focus detection after extraction? Merely an assumption?
The false negative result has been proven by microbiological detection after extraction, not by an assumption, because the patient underwent surgical CIED removal (new version page 9, lines 267-270).
- Discussion: line 306: “left vs right ventricle” – I don’t understand what you mean by this, please elaborate.
Thank you for this suggestion; the statement has been adjusted explicating that “left vs right” was referred to blood shunt (new version page 9, line 277).
- Discussion: If a direct comparison in SUVmax and SQR is not possible between studies, what is the value of reporting them? Please elaborate further.
We made a mistake in reporting the sentence in this way, so we modified it, specifying that the comparison is possible but can be interfered (new version page 9, lines 299-300).
- Discussion: It should be explained that FDG PET/CT is more accurate in pocket infection than it is in lead infection, and that the high levels of sensitivity found are probably due (at least in part) to the high number of pocket infections in this cohort.
Thank you for this suggestion; the reviewer statement has been introduced in the main text (new version page 9, lines 273-274).
- Conclusions: “It can help the clinicians in decision making, (…)” – This leads me back to question 1: How can you conclude FDG PET/CT can help in decision making when your study design explicitly did not let the PET/CT results influence clinical decision making?
Our conclusions are related to the rational of our study as reported in the answer to the first reviewer’s comment and the goal of our study was to provide further results in support or not of the usefulness of PET/CT in this field (new version page 2, lines 55-60; references #15 to #19; new version page 11, lines 378-393).
Round 2
Reviewer 2 Report
I thank the authors for making adjustments to their paper and addressing most of my concerns.
I must continue to dispute the retrospective nature of this study if PET-scans were performed knowing that they would not impact clinical decision making. I will defer to the judgment of the editor in this case, as I have made my reservations clear regarding this point.
Regarding figures 3&4, I would like to meet the authors half-way, so to speak: Please add a figure with the data points for the three groups (e.g. like figure 5 from reference 21), i.e. positive (11/15), negative (4/15) and controls (15/15) for both SUVmax and SQR as a new figure 3, and bring the current figures 3&4 together in a single new figure 4.
This way, the raw data is available as well as the analysis as performed by the authors.
Thank you, again, for the opportunity to review your paper.
Author Response
- I must continue to dispute the retrospective nature of this study if PET-scans were performed knowing that they would not impact clinical decision making. I will defer to the judgment of the editor in this case, as I have made my reservations clear regarding this point.
This dispute is positive and constructive; we understand the point of view of the reviewer nevertheless we continue to support ours; anyway at the state of art a prospective analysis needs the ethical committee approval by our institution and obtaining it means loosing the possibility to publish these data. We hope it will not happen.
- Regarding figures 3&4, I would like to meet the authors half-way, so to speak: Please add a figure with the data points for the three groups (e.g. like figure 5 from reference 21), i.e. positive (11/15), negative (4/15) and controls (15/15) for both SUVmax and SQR as a new figure 3, and bring the current figures 3&4 together in a single new figure 4.
This way, the raw data is available as well as the analysis as performed by the authors.
We would like to report a heartfelt thanks to the reviewer for this suggestion; we have inserted the figure that he suggests with the relative paragraph in the text (new version page 7, lines 211-220) and the current version satisfies us very much because this reviewer solution enhances our work. Thanks again.